# Risk Stratification of Thyroid Nodules: From Ultrasound Features to TIRADS

**DOI:** 10.3390/cancers14030717

**Published:** 2022-01-30

**Authors:** Teresa Rago, Paolo Vitti

**Affiliations:** Department of Clinical and Experimental Medicine, Endocrinology Section, University of Pisa, 56124 Pisa, Italy; paolo.vitti@med.unipi.it

**Keywords:** thyroid nodule, thyroid cancer, ultrasonography, elastosonography, fine-needle aspiration

## Abstract

**Simple Summary:**

Thyroid nodules are a frequent clinical issue. Their incidence has increased mainly due to the widespread use of neck ultrasound scans. Most thyroid nodules are asymptomatic, incidentally discovered, and benign at cytology. Thyroid ultrasound is the most sensitive diagnostic tool to evaluate patients with nodular thyroid disease. It is therefore important to use the ultrasound features to select nodules that require a fine-needle aspiration cytology.

**Abstract:**

Thyroid nodules are common in iodine deficient areas, in females, and in patients undergoing neck irradiation. High-resolution ultrasonography (US) is important for detecting and evaluating thyroid nodules. US is used to determine the size and features of thyroid nodules, as well as the presence of neck lymph node metastasis. It also facilitates guided fine-needle aspiration (US-FNA). The most consistent US malignancy features of thyroid nodules are spiculated margins, microcalcifications, a taller-than-wide shape, and marked hypoechogenicity. Increased nodular vascularization is not identified as a predictor of malignancy. Thyroid elastosonography (USE) is also used to characterize thyroid nodules. In fact, a low elasticity of nodules at USE has been related to a higher risk of malignancy. According to their US features, thyroid nodules can be stratified into three categories: low-, intermediate-, and high-risk nodules. US-FNA is suggested for intermediate and high-risk nodules.

## 1. Introduction

Thyroid nodules are detected in 50–65% of healthy individuals, the majority being asymptomatic and discovered incidentally [1,2]. Most are benign and do not require treatment [1,2]—less than 5% being malignant. Thyroid nodules are more common in iodine deficiency areas, in females, and in patients undergoing neck irradiation. In rare cases, a thyroid nodule can cause compressive symptoms or hyperthyroidism, thus requiring treatment.

The risk factors associated with a higher probability of malignancy include a history of neck irradiation, a family history of medullary thyroid carcinoma or multiple endocrine neoplasia (MEN2), age < 20 years or >60 years, male sex, rapid growth, a firm and hard consistency, and the presence of suspicious cervical lymph nodes [3,4,5,6,7,8].

US is the most important diagnostic tool for detecting thyroid nodules [1,2,9,10]. In addition, US can be used to determine the size and features of palpable and nonpalpable nodules, to guide fine-needle aspiration (FNA), and to diagnose lymph node metastasis. 

Although thyroid US has been considered as the cornerstone for the management of thyroid nodules, there is no clear consensus on nodule selection for US-guided FNA [11,12,13,14], on a standardized terminology for US features [15,16,17,18,19,20]. Due to their increased detection, thyroid nodules represent a clinical challenge [15,16,17,18,19,20]. Initial evaluation should include physical examination and investigation of risk factors, such as previous radiation exposure, family history of thyroid diseases, lump growth rate, and signs and symptoms of compression [1,2]. When there is a suspicion that a nodule is functioning (i.e., low TSH), thyroid scintigraphy is mandatory. US examination should always be recommended [1,2]. Several endocrine societies have developed various US-based guidelines and recommendations for managing thyroid nodules [16,17,18,19,20,21,22].

## 2. Real-Time US Findings of Thyroid Nodules

*Size*: Although the size of a thyroid nodule is not helpful in predicting malignancy, the size should be measured in all three dimensions and recorded for follow-up. Malignant nodules grow faster than benign nodules, but 90% of the latter grow by 15% during a 5-year follow-up period [23,24,25]. Cystic nodules show a slower growth than solid nodules [25]. Sudden growth of solid nodules may be a clinical manifestation of high-grade malignancy, such as anaplastic thyroid carcinoma or lymphoma [1,2].

There is no clear consensus on the definition of nodule growth. According to the American Thyroid Association (ATA) guidelines, a definition of growth is a 20% increase in the diameter of the nodule with a minimum 2 mm increase in 2 diameters [1]. Some authors prefer a 15% increase in lump volume as a definition of nodule growth [25,26]. However, inter-observer variability has been observed in small nodules, especially for a volume increase of less than 50%. Nodule growth is therefore defined as a 20% increase in diameter or a 50% increase in volume [1,27,28].

*Aspect:* Thyroid cancer is rare in cystic nodules, although 13–26% of thyroid cancers may have a cystic component [29,30]. Rarely, partially cystic nodules can be malignant [31]. In this case, papillary thyroid carcinoma may have an eccentric solid component with vascularization, or the presence of microcalcifications [29,31,32]. A lump is called spongiform when a microcystic appearance occupies more than 50% of the lump and is considered a sign of benignity with high specificity [33,34,35]. A nodule can be classified according to the ratio between the solid component and the cystic one as: solid (≤10% of the cystic portion), predominantly solid (from >10% to ≤50% of the cystic portion), predominantly cystic (from >50% to ≤90% of the cystic portion), and cystic (>90% of the cystic portion) [33].

*Shape:* The shape of a nodule has gained diagnostic importance for the differentiation between benign and malignant nodules since the first observation by Kim et al. [36,37,38], who reported that a taller-than-wide shape had 93% specificity for diagnosing malignancy. In a larger multicenter study, a taller-than-wide shape was shown to be highly suggestive of malignancy with a specificity of 89% and a positive predictive value of 86% [33]. These results are explained by the growth pattern, because malignant nodules grow through the normal tissue plane in a centrifugal way, while benign nodules grow along the tissue plane in a parallel fashion [36,37,38,39]. In benign nodules, the shape can therefore be ovoid to round (the antero–posterior diameter is less or equal to its transverse diameter on a transverse plane).

*Halo sign*: Nodules may have a thin or thick halo. A halo or hypoechoic rim surrounding a nodule consists of a pseudocapsule due to fibrous connective tissue, compressed thyroid tissue, and chronic inflammatory process [40]. Although a completely uniform halo is suggestive of benignity with a specificity of 95% [41], more than half of benign nodules are devoid of a halo [30,31,32,33,34,35,36,37,38,39,40]. An uneven thick or incomplete halo due to a fibrotic pseudocapsular structure and inflammatory and necrotic process is observed in 10–12% of papillary thyroid carcinomas and is frequently associated with an irregular shape. On the other hand, 10–24% of papillary carcinomas have a complete or incomplete halo [29,30,41].

*Margins:* Previous studies have reported that both spiculated or microlobulated margins and poorly defined margins are suggestive of malignancy [36,42]. Nodule margins are ill-defined when they lack clear demarcation from the surrounding perinodular tissue for most of (>50%) their edge [33].

When the tumor infiltration of the margin is minimal, it manifests as an ill-defined margin. However, benign thyroid nodules are sometimes incompletely encapsulated and poorly marginated, and they can merge with normal tissue [43]. Therefore, an ill-defined margin is a nonspecific finding that can be observed in both benign and malignant nodules. Conversely, a spiculated margin is highly suggestive of malignancy with a specificity of 92% and a positive predictive value of 81% [33]. We thus suggest that the margin of a nodule is classified as follows: smooth, spiculated/microlobulated, or ill-defined.

*Echogenicity:* Marked hypoechogenicity is highly specific for malignancy with a specificity of 92–94% [33,36]. Although the thyroid parenchymal echogenicity is different in different individuals, it is used as a reference for nodule echogenicity. Neck strap muscles (the sternothyroid, sternocleidomastoid), characterized by very low echogenicity, are also used as a reference tissue [33,36]. The salivary glands may also be used as a standard of normal thyroid echogenicity in patients with hypoechogenicity. Nodule echogenicity is classified as follows: marked hypoechoic (nodule echogenicity is similar to that of the adjacent neck strap muscles), hypoechoic, isoechoic, or hyperechoic, compared with the echogenicity of the normal thyroid parenchyma.

*Calcifications:* A calcification is defined as an echogenic focus with or without back shadow. The absence of posterior shadow does not exclude calcification as some calcifications are too small to produce a posterior shadow. Punctate echogenic foci with reverberation artifacts are due to colloid materials and can be easily differentiated from calcifications on US. Some studies report that all types of calcifications seen on US increase the likelihood of malignancy. In particular, comet-tail artifacts can represent dense colloid, fibrin deposits and even microcalcifications. The presence of comet-tail artifacts in a cystic nodule is highly suggestive of benignity but may not rule out malignancy if present in a solid component. Moreover, the punctate echogenic foci do not necessarily represent the psammoma bodies that are observed in papillary thyroid carcinoma but may be dystrophic calcifications or microdeposit of dense colloid.

Calcifications can be observed in both benign and malignant nodules. We suggest classifying calcifications as follows: (i) microcalcifications—small dotted echogenic foci of 1 mm or less either with or without posterior shadow; (ii) macrocalcifications—dotted echogenic foci larger than 1 mm; (iii) coarse or peripheral and border calcifications.

At histology, microcalcifications correspond to psammoma bodies, which are round, laminar, crystalline, calcific deposits 10–100 μm, specific to papillary thyroid carcinoma. Microcalcifications are highly suggestive of malignancy with a specificity of 86–95% and a positive predictive value of 42–94% [33,36,42,44,45]. Large and irregular shaped dystrophic calcifications may be due to tissue necrosis and can be observed in benign and malignant nodules. The significance of peripheral, eggshell, or rim calcification is still debated in terms of differentiation between benign and malignant nodules. In longstanding hyperplastic nodules peripheral rim calcification may be present. However, the focal disruption of the eggshell structure associated with the presence on a thick and markedly hypoechoic halo can be predictive of malignancy [34,46,47,48].

## 3. Accessory Features

*US in Lymph nodes*: US examen of the cervical lymph nodes should be performed in all patients with thyroid nodules. The US appearance of a typical normal lymph node is hypoechogenicity, an oval shape and presence of the central hyperechoic streak corresponding to the hilum. On the other hand, a pathological lymph node can be cystic or solid, iso or hyperechoic, round or irregular in shape, and without the hilum [49]. The position of the described lymph nodes must be precisely located following Robbins’ scheme [50]. In suspicious lymph nodes, an US-guided FNA should be performed for cytology and thyroglobulin or calcitonin measurement in the needle washout.

*Extrathyroidal extension (EE):* EE is characterized by protrusion into adjacent structures and/or rupture of the capsular margin of the thyroid neoplasm. In small tumors, the presence of EE is very important in deciding the type of surgery: lobectomy versus total thyroidectomy. The presence of minimal EE is not associate to a worse prognosis of the tumor. The US features that define EE are contact, degree of contact, and interruption of the capsule. Kwak reports that a greater than 25% contact between the thyroid nodule and the adjacent capsule is a useful US marker for predicting the presence of EE [51]. Capsular abutment has less specificity. On the other hand, the presence of more than 2 mm normal thyroid parenchyma between the nodule and a continuous capsule reduces the risk of microscopic extrathyroidal extension to less than 6% [51,52,53,54].

## 4. Color Doppler Flow Imaging (CDFI), Power Doppler US, and Superb Micro-Vascular Imaging (SMI)

US color Doppler or US power Doppler provide information on the vascularization of the nodules. Although vascularity is a nonspecific finding in thyroid cancer, it is found in 69–74% of cases [29]. Benign nodules are characterized by a perinodular flow which, however, can also be observed in 22% of carcinomas [29]. Intranodular vascularization is observed in carcinoma but has a low specificity, while chaotic vascularization is more specific, but with a very low sensitivity [35]. Some studies report that the resistance index, maximum systolic velocity, and vascularization pattern on a Doppler US do not differ between benign nodules and carcinomas [55,56,57]. Color and power Doppler only provide complementary information and are even less reliable for small nodules (<5 mm) due to the misinterpretation of perinodular vessels as an intranodular vascular signal. Therefore, several authors advise against the routine use of color and power Doppler US for thyroid nodules [1].

CDFI uses low-frequency, low-speed flow signals, while contrast-enhanced ultrasound (CEUS) detects low-frequency flow signals with a diameter of 10–30 μm and a flow rate of approximately 1 mm/s. CEUS is expensive and can cause an allergic reaction [58,59,60,61]. SMI is a recently introduced, non-invasive, inexpensive exam that highlights microflows and detects tissue signals, thus minimizing artifacts. There are few data in the literature on the usefulness of this investigation for the characterization of thyroid nodules [61].

In CEUS analysis, a high perfusion indicates an extensive microvasculature, whereas a low perfusion suggests a lower degree of microvasculature. Some reports have shown that malignant nodules had mainly hypo-enhancement [62,63,64,65,66], which can be due to fibrosis and neovascular damage by tumor cells, while benign nodules had hyper-enhancement or iso-enhancement, similar to normal tissue. Zang et al. reported a higher sensitivity and lower specificity of CEUS + SMI in 75 suspicious nodules, compared with CEUS or SMI alone [67].

## 5. US Elastosonography (USE)

USE determines the elasticity of tissue. Given that a carcinoma is harder than a normal thyroid parenchyma or a benign nodule, a high stiffness on USE has been suggested as a good predictor of malignancy [68,69,70,71,72,73]. Our group showed that low elasticity scores, indicative of a hard consistency, were associated with malignancy with a specificity of 100% and sensitivity of 97% [69]. The predictivity of the USE measurement was independent of the nodule size. In fact, a high sensitivity and specificity were found even in nodules with the largest diameter of 0.8–1 cm. In a large series of patients with indeterminate and non-diagnostic cytology, our group confirmed that high nodular stiffness is associated with malignancy. In this paper, we also simplified the classification of USE into 3 groups: score 1—nodules with uniform high elasticity, probably benign; score 3—nodules with uniform low elasticity, probably malignant; score 2—nodules with a non-homogeneous distribution of elasticity, suspicious. Since the vast majority of nodules with indeterminate and non-diagnostic cytology had a score of 1, they had a low probability of malignancy [74]. Our findings may limit the indications for surgical treatment to the subgroup of patients with the highest risk of cancer. We also showed that in 115 patients who underwent surgery for a suspicious cytology, or large nodules with suspicious US features and non-diagnostic cytology, low elasticity at USE was highly correlated with malignancy and also with the presence of fibrosis and expression of Gal-3 and FN-1 in the histological specimens [75]. A few pitfalls limit the diagnostic usefulness of USE, which is operator dependent. Moreover, cystic lesions, nodules with calcified shell and multinodular goiter with coalescent nodules are not suitable for USE evaluation.

## 6. US Risk Stratification Systems: The TIRADS

US-guided FNA is the main diagnostic tool for detecting malignancy in thyroid nodules. Its use should be restricted to thyroid nodules suspicious for malignancy. In fact, most scientific societies agree that US features should support the indications to perform US-guided FNA. Several classification systems have been proposed aimed at stratifying the risk of cancer in thyroid nodules [17,18,19,20,21].

However, apart from the well-recognized advantage, thyroid US also has drawbacks such as the poor reproducibility, due to the different equipment used, lack of a standardized US report and inter- and intra-operator variability. To address these main points, several US risk stratification systems (i.e., thyroid imaging reporting and data systems—TIRADS) have been developed to stratify the malignancy risk of a nodule and then suggest the need for US-guided FNA [76,77,78,79,80,81,82]. These systems are called TIRADS because they were modeled in line with the American Committee of Radiology BIRADS, which has been widely accepted in breast imaging.

The TIRADS classification is a point scale that categorizes the US of thyroid nodules from low to high suspicion, based on the number and combination of the predictors of malignancy [17]. Initially, Horvath et al. in 2009 [17] proposed a classification system, which assigned levels of malignancy risk to different patterns, involving 10 features. On the other hand, Park et al. devised an equation to predict the probability of malignancy based on 12 variables. Kwak proposed a simplified system in which nodules were stratified only on the basis of five US patterns [18].

So far, many professional societies have proposed US-based risk stratification systems. The Chinese-TIRADS was recently proposed from Chinese professional society and the revised 2021 Korean-TIRADS was very recently published [19,20]. The TIRADS classifications have been slightly modified over the years and different versions have been suggested by different guidelines, including EU-TIRADS provided by the European Thyroid Association [79], ACR-TIRADS by the American College of Radiology [22], and K-TIRADS by the Korean Society of Thyroid Radiology [20,21]. These different versions of TIRADS have been validated and have shown great diagnostic value in predicting thyroid malignancy. However, most of those studies were retrospective and the results heterogeneous, limiting their applicability in clinical practice. In a recent meta-analysis, Castellana et al. assessed the prevalence of malignancy in each EU-TIRADS, class 5 compared to classes 2, 3, and 4. The authors found that the prevalence of malignancy was 16% in class 2, 5.5% in 3, 20.6% in 4, and 83.3% in 5 [83,84]. These findings were very close to the estimates of the ETA experts. EU-TIRADS should therefore be considered as an accurate way of stratifying the risk of malignancy of thyroid nodules and performing US-guided FNA is not recommended in EU-TIRADS class 2 nodules. However, the risk of malignancy is greater in highly specialized centers than in primary care centers. This is linked to the fact that selected patients come to highly specialized centers. This explains why in EU-TIRADS there is an overestimation of the risk of malignancy.

A recent consensus of the Italian Thyroid Association, the Italian Society of Endocrinology, the Italian Society of Ultrasonography in Medicine and Biology, and the Ultrasound Chapter of the Italian Society of Medical Radiology considered that the main limitation of US is the poor reproducibility, due to the varying experience of the operators and the different performance and settings of the equipment. A simplified nodule risk stratification was therefore proposed, which is based on the predictive value of each US sign, classified and evaluated according to the strength of association with malignancy, but also to the estimated reproducibility between different operators [85]. The risk score was classified into four categories on the basis of the estimated specificity and reproducibility among different operators for each US feature (Table 1). The risk score is the sum of the single scores attributed to each US pattern.

## 7. Indication for FNA According to US Risk Stratification Systems

As noted above, the recommendation as to whether or not to perform US-guided FNA depends on US features associated with malignancy, size, and patient’s history. US-FNA has a high sensitivity in small nodules with suspicious US features, while in large nodules, the sensitivity is reduced. Furthermore, considering that the prognosis of some tumors (such as follicular or Hurthle cell carcinoma) is related to the size of the nodule, it is important to recommend US-FNA in nodules > 2 cm in size, or in those that grow over time. Thus, most guidelines recommend FNA in solid nodules > 2 cm even when devoid of US signs suggestive of malignancy. A point of discussion is the size below which FNA is not indicated. In fact, the mortality and recurrence rate of thyroid cancer is directly proportional to the size of the nodules [1,2,19]. The ATA and ETA guidelines recommend US-guided FNA in sub-centimetric nodules only when suspicious features are present and in patients with a history of radiation exposure or familial thyroid cancer [1,2]. Thyroid carcinoma smaller than 5 mm compared with 6–10 mm diameter has a better survival and less recurrence at 5 years (<3% versus 14%) [1]. Recent studies thus recommend not performing US-guided FNA (Table 2) in nodules smaller than 5 mm, also due to the high rate of false positive US findings and the high rate of inadequate cytology [8].

In nodular goiter, US-guided FNA cannot be performed in all nodules. The risk of malignancy for patients with multiple thyroid nodules is not very different from the risk for patients with a single thyroid nodule [9,86]. According to the guidelines [1,2,78], in the presence of 2 or more nodules equal or greater than 1–1.5 cm, US-guided FNA is recommended for those with suspicious US features. If none of the nodules have suspicious US features, FNA of the largest nodule should be performed. All the guidelines agree that US-guided FNA should be advised in high and intermediate risk category nodules, and not the low-risk category.

In summary, the three main aims of using US risk stratification systems are the following: (i) to contribute to the optimal management strategy; (ii) to reduce the number of unnecessary investigations; (iii) to select those patients who should be operated on. The secondary objectives are to facilitate communication between professionals and patients, facilitate a cross-dialogue between clinicians and pathologists, and improve the inter-observer agreement of US reports.

## 8. Conclusions

Today thyroid nodules are frequently detected by imaging techniques. Only a minority of these nodules will cause significant harm to health. Thyroid US is primarily responsible for this frequent detection, and is also the primary tool for stratifying the risk of cancer and the strength of US-guided FNA indication.

## Figures and Tables

**Table 1 cancers-14-00717-t001:** Stratification of the risk score based on the predictive value of each US feature associated with malignancy.

US Features Associated with Malignancy	Low Specificity/High Reproducibility	Hypoechogenicity Thick Halo	Score Value 1
	**High specificity/poor reproducibility**	Microcalcifications irregular, disrupted, spiculated or lobulated margins, high stiffness at USE	**Score Value 2**
	**High specificity/high reproducibility**	Marked hypoechogenicity irregular shape, taller-than-wide	**Score Value 3**
	**Very high specificity/high reproducibility/Accessory features**	Extracapsular extension, suspicious lymph nodes	**Score Value 4**
**Risk category**	**1. Low risk**	Nodules with at least 2 US features associated with benignity * and no features associated with malignancy	
	**2. Intermediate risk**	Nodules with total risk score 1–3	
	**3. High risk**	Nodules with total risk score ≥4	

Modified by Rago et al. [85]. The risk score is the sum of the single scores attributed to each ultrasound feature. * Purely cystic nodules, mixed nodules with liquid content, spongiform nodules, oval shape, isoechoic/hyperechoic nodules with complete halo sign, isoechoic/hyperechoic nodules with complete halo sign and lamellar macrocalcifications, hyperechoic pseudonodular areas in thyroid autoimmune diseases.

**Table 2 cancers-14-00717-t002:** US risk stratification for malignancy and indication for US-FNA.

	French [77] 2013		ATA [1] 2016		ACE/ACE-AME [2] 2016		Korean [20] 2021		ETA [79] 2018	
Risk Category	M.R. (%)	FNA Size (cm)	M.R. (%)	FNA Size (cm)	M.R. (%)	FNA Size (cm)	M.R. (%)	FNA Size (cm)	M.R. (%)	FNA Size (cm)
**High**	100	≥1	70–90	>1	50–90	≥1	>60	>1–1.5	26–87	>10
**Intermediate**	69	≥1	10–20	≥1	5–15	>2	15–40	>2	6–17	>15
**Low**	6	≥1.5	5–10	≥1.5	1	≥2	3–10	≥1.5	2–4	>20
**Very low**	0.25	≥2	≥3	>2			<3			
**Benign**	0	NA	<1	NA				NA	0	NA
**No Nodule**										

M.R.—malignancy risk; N.A—not advised.

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
