# Peer review of "Risk Stratification of Thyroid Nodules: From Ultrasound Features to TIRADS"

_cancers, 2022, doi:10.3390/cancers14030717_

Round 1

Reviewer 1 Report

General comments:

Authors comprehensively reviewed the current status and issues of US lexicons and US-based risk stratification of thyroid nodules. This review is well-organized and has advantages of focusing on clinically important issues in US evaluation and management of patients with thyroid nodules. However, it is uncertain whether this review highlights the updated knowledge and perspective on the relevant issues compared to recently published similar review articles on similar issues, and the majority of the cited research references are rather old and need to be updated.

Specific comments

  1. So far, many professional societies have proposed US-based risk stratification systems. The Chinese-TIRADS was recently proposed from Chinese professional society (2020 Chinese guidelines for ultrasound malignancy risk stratification of thyroid nodules: the C-TIRADS. Endocrine 2020;70:256-279) and the revised 2021 Korean-TIRADS was very recently published (2021 Korean Thyroid Imaging Reporting and Data System and Imaging-Based Management of Thyroid Nodules: Korean Society of Thyroid Radiology Consensus Statement and Recommendations. Korean J Radiol. 2021 Oct 26), which need to be included for the updated review.
  2. There are many controversial issues on the US lexicons regarding the definition of US terminologies and no established standardized US lexicons. There are also conflicting study results on the predictive value of several US features for malignancy. The comparison of definition and descriptors of US lexicons used for the risk stratification among the proposed professional societies’ RSSs (guidelines) will be useful for the comprehensive review on the controversial issues in US lexicons. For, example, the punctate echogenic foci (microcalcification) has been considered as a specific predictor for malignancy, however, the predictive value of suspicious US features such as microcalcification and taller-than-wide shape strongly depends on the composition and echogenicity of nodules based on recent studies. Although the echogenic foci with comet tail artifact (one of reverberation artifact) has been considered as a predictor for benign nodules, recent studies consistently shows that it is not a predictor for benignity in solid and hypoechoic nodules and occasionally found in PTC.
  3. Although authors introduced the risk stratification system proposed by Italian societies, deep analysis of differences and similarities among the RSSs proposed by professional societies would help readers to understand the current issues on US-RSSs for thyroid nodules and get the further perspective towards developing the unified international RSS.
  4. There are errors and inappropriate citations in some parts of the manuscripts. The appropriateness of cited references should be checked throughout the manuscript.

- P2, “Several endocrine societies have developed various US-based guidelines and recommendations for managing thyroid nodules [16-20].” The cited references are proposals by researchers (16-18) and by radiology societies (19, 20), not by endocrine societies.

- P2. Shape “In a larger multicenter study, a taller-than-wide shape was shown to be highly suggestive of malignancy with a specificity of 89% and a positive predictive value of 86% [34].” The cited reference is a single center study for 155 nonpalpable small nodules and the result data are different.

Author Response

The authors are  grateful to to the Referee 1 for his/her kind remarks and constructive criticism 

  1. So far, many professional societies have proposed US-based risk stratification systems. The Chinese-TIRADS was recently proposed from Chinese professional society (2020 Chinese guidelines for ultrasound malignancy risk stratification of thyroid nodules: the C-TIRADS. Endocrine 2020;70:256-279) and the revised 2021 Korean-TIRADS was very recently published (2021 Korean Thyroid Imaging Reporting and Data System and Imaging-Based Management of Thyroid Nodules: Korean Society of Thyroid Radiology Consensus Statement and Recommendations. Korean J Radiol. 2021 Oct 26), which need to be included for the updated review. These papers were quoted (page 8 second paragraph, Ref 19-20)
  2. Thanks to Referee 1 There are many controversial issues on the US lexicons regarding the definition of US terminologies and no established standardized US lexicons. There are also conflicting study results on the predictive value of several US features for malignancy. The comparison of definition and descriptors of US lexicons used for the risk stratification among the proposed professional societies’ RSSs (guidelines) will be useful for the comprehensive review on the controversial issues in US lexicons. For, example, the punctate echogenic foci (microcalcification) has been considered as a specific predictor for malignancy, however, the predictive value of suspicious US features such as microcalcification and taller-than-wide shape strongly depends on the composition and echogenicity of nodules based on recent studies. Although the echogenic foci with comet tail artifact (one of reverberation artifact) has been considered as a predictor for benign nodules, recent studies consistently shows that it is not a predictor for benignity in solid and hypoechoic nodules and occasionally found in PTC.

The authors agree that terminology is not uniform and many conflicting results are reported on the predictivity of single US features, but discussion of controversies present in the literature on single patterns would be hard to report in a review. Thus, only some of these discrepancies are discussed in the text (see pag 7 second paragraph  6. US Risk stratification systems). This topic is discussed in the paper (Ref. 85) by Rago et al. and indeed this was the exactly the aim of this paper, in which the proposal was done to give a different weight to each US sign accordingly not only to the strength of association with malignancy, but also to the estimated reproducibility among different operators. This point has been better clarified in the revised version in pag 8 last  paragraph.

Thus, some studies report that all types of calcifications seen on US increase the likelihood of malignancy. In particular, comet-tail artifacts can represent dense colloid, fibrin deposits and even microcalcifications. The presence of comet-tail artifacts in a cystic nodule is highly suggestive of benignity, but may not rule out malignancy if present in a solid component. Moreover, the punctate echogenic foci do not necessarily represent the psammoma bodies that are observed in the PTC but may be dystrophic calcifications or microdeposit of dense colloid. Therefore, with ultrasound it can empirically establish if the punctate echogenic foci are microcalcifications or dense colloid. So in many cases it is difficult to define what punctate echogenic foci are.

In the RV this point was added on page 4 section “calcification” line 5

  1. Although authors introduced the risk stratification system proposed by Italian societies, deep analysis of differences and similarities among the RSSs proposed by professional societies would help readers to understand the current issues on US-RSSs for thyroid nodules and get the further perspective towards developing the unified international RSS.

The authors agree on this point. A Table is modified..

  1. There are errors and inappropriate citations in some parts of the manuscripts. The appropriateness of cited references should be checked throughout the manuscript.- P2, “Several endocrine societies have developed various US-based guidelines and recommendations for managing thyroid nodules [16-20].” The cited references are proposals by researchers (16-18) and by radiology societies (19, 20), not by endocrine societies.  P2. Shape “In a larger multicenter study, a taller-than-wide shape was shown to be highly suggestive of malignancy with a specificity of 89% and a positive predictive value of 86% [34].” The cited reference is a single center study for 155 nonpalpable small nodules and the result data are different.

Thanks to Referee, the author agree the correct reference  is 33  and  modified in RV

Parts that have been modified are highlighted in yellow characters.

Authors hope that the manuscript is now acceptable for publication in Cancer .

Reviewer 2 Report

General:

The review is adequately tailored to the topic. In my opinion, the main issue on TIRADS is a statistical one: TIRADS was derived from datasets with an a priori risk for thyroid cancer of about 15% (e.g. Horvath, Kwak…). However, in primary care units the a priori risk for cancer in an individual thyroid nodule ist about 1 to 5 per mille, i.e. two orders of magnitude lower. For this reason, according to the Bayes theorem for conditional probabilities, the TIRADS ultrasound features – in particular the recommendations for FNA – work only at institution with a comparable a priori risk like in the initial studies, i. e. tertiary or quartary care units. For primary care units it does by no means work but would yiel far too much doubtful FNA results and thus futile diagnostic operations. Honestly, so far, no appropriate TIRADS has been published for primary care units.

I would appreciate very much if the authors could address this issue. Along with this issue I want to remind the fact that, in primary care units, thyroid scintigrams are necessary to exclude hot nodules from furhter diagnostic work up as given in:

Risk Stratification of Thyroid Nodules Using the Thyroid

Imaging Reporting and Data System (TIRADS): The Omission

of Thyroid Scintigraphy Increases the Rate of Falsely

Suspected Lesions

Simone Schenke1, Philipp Seifert2, Michael Zimny1, Thomas Winkens2, Ina Binse3, and Rainer G¨orges3,4

The aspect of hot nodules has been widely omitted in all TIRADS-system for the above mentioned reasons but should be included in this review. In primary care units it might also be useful to apply TIRADS in a very restrictive way in order to not include too many nodules – but include solid nodules which are „cold“ on scintigrams. This is routine at least in many epidemic areas such as Germany. For this reason your suggestion of diagnostic work-up (table 1) should be modified to  he effect that it largely depends on the a priori risk (i. e. level of care), local circumstances and availabilities (e.g. availability of thyroid scans)

Details

Simple Summary: ok

Abstract: wording, sometimes hard to unterstand („if such features ar not recognized, increased nodular vascularization is not identified as a predictor of malignancy“ ?!).

Intro: wording

  1. 2. wording

„Malignant nodules grow faster than benign ones…“ this is true only for anaplastic carcinoma.

Approximately 5% of all partially cystic nodules have been reported to be malignant (19)“ – not true in primary care units. These absolute values for malignancy risks are misleading since they are highly dependent on the a priori risk for thyroid cancer in a given patient group. For this reason this number should not be cited.

The US appearance of a typical lymph node is – please add: typical normal lymph node

EE is characterized by protrusion into adjacent struc-tures and / or rupture of the capsular margin of the neoplasm“.thyroid, not neoplasm

The presence of EE leads to a worse prognosis of the tumor. This is a matter of debate, According to UICC 2017 (TNM) minimal EE is no longer considere a risk factor.

Kwak reports that greater than 25% contact between the thyroid nodule and the adjacent capsule is a useful US marker for predicting the presence of EE [18]. – i couldn’t find the according data in the cited publication. The cited publication is about the initial „Kwak“ system but not on EE. Maybe it is another publication from Kwak?

Castellana et al. assessed the prevalence of malignancy in each EU-TIRADS, class 5 compared to classes 2, 3 and 4. The authors found that the prevalence of malignancy was 16% in class 2, 5.5% in 3, 20.6% in 4 and 83.3% in 5 [82]. The studies included in that review show malignancy risks for thyroid carcinom around 50% - which is in no way comparable to primary care where this risk is far below 1%. For this reason the EU-TIRADS greatly overestimates malignancy risks – should be discussed. Bayes theorem.

Table 1: not readily understandable: must the scores be summed?

The ATA and ETA guide-lines recommend US guided FNA in subcentimetric nodules only when suspicious fea-tures are present – somewhat misleading, since the US signs are not meant in ATA, but factors like suspicion of lymph node metastases, ETE and so on.

Author Response

The authors are  grateful to to the Referee 2 for his/her kind remarks and constructive criticism

  1. The review is adequately tailored to the topic. In my opinion, the main issue on TIRADS is a statistical one: TIRADS was derived from datasets with an a priori risk for thyroid cancer of about 15% (e.g. Horvath, Kwak…). However, in primary care units the a priori risk for cancer in an individual thyroid nodule is about 1 to 5 per mille, i.e. two orders of magnitude lower. For this reason, according to the Bayes theorem for conditional probabilities, the TIRADS ultrasound features – in particular the recommendations for FNA – work only at institution with a comparable a priori risk like in the initial studies, i. e. tertiary or quarterly care units. For primary care units it does by no means work but would yiel far too much doubtful FNA results and thus futile diagnostic operations. Honestly, so far, no appropriate TIRADS has been published for primary care units. I would appreciate very much if the authors could address this issue. Along with this issue I want to remind the fact that, in primary care units, thyroid scintigrams are necessary to exclude hot nodules from further diagnostic work up as given in: Risk Stratification of Thyroid Nodules Using the Thyroid Imaging Reporting and Data System (TIRADS): The Omission of Thyroid Scintigraphy Increases the Rate of Falsely Suspected Lesions Simone Schenke1, Philipp Seifert2, Michael Zimny1, Thomas Winkens2, Ina Binse3, and Rainer G¨orges3,4The aspect of hot nodules has been widely omitted in all TIRADS-system for the above mentioned reasons but should be included in this review. In primary care units it might also be useful to apply TIRADS in a very restrictive way in order to not include too many nodules – but include solid nodules which are „cold“ on scintigrams. This is routine at least in many epidemic areas such as Germany. For this reason your suggestion of diagnostic work-up (table 1) should be modified to the effect that it largely depends on the a priori risk (i. e. level of care), local circumstances and availabilities (e.g. availability of thyroid scans)

The authors agree, however in the primary care units clinical evaluation is important to set up the subsequent diagnostic framework which includes ultrasound and cytological examination on FNA and  according to the Referee’s suggestion  we have added 2 sentences on the issue of primary care units (Page 2 line 18) and on thyroid scintigraphy ( page 2 line 19)

Simple Summary: ok

  1. Abstract: wording, sometimes hard to unterstand („if such features ar not recognized, increased nodular vascularization is not identified as a predictor of malignancy“ ?!).

The authors agree and the sentence “if such features are not recognized”, is deleted

  1.  Wording “Malignant nodules grow faster than benign ones…“ this is true only for anaplastic carcinoma.

The author agree, the sentence is modified in page 2 paragraph Size line 6.

  1. Approximately 5% of all partially cystic nodules have been reported to be malignant (19)“– not true in primary care units. These absolute values for malignancy risks are misleading since they are highly dependent on the a priori risk for thyroid cancer in a given patient group. For this reason this number should not be cited. The author agree, the sentence is modified in page 3 paragraph “Aspect” line 2

  1. The US appearance of a typical lymph node is– please add: typical normal lymph node.

The author agree, the sentence is modified as the Referee’s suggestion

  1. EE is characterized by protrusion into adjacent struc-tures and / or rupture of the capsular margin of the neoplasm“.– thyroid, not neoplasm

The author agree, the sentence is modified  as Referee’s suggestion

  1. The presence of EE leads to a worse prognosis of the tumor. This is a matter of debate, According to UICC 2017 (TNM) minimalEE is no longer considered a risk factor.

The author agree, the sentence is modified  as the Referee’s suggestion page 5 section “EE”

  1. Kwak reports that greater than 25% contact between the thyroid nodule and the adjacent capsule is a useful US marker for predicting the presence of EE [18].– i couldn’t find the according data in the cited publication. The cited publication is about the initial „Kwak“ system but not on EE. Maybe it is another publication from Kwak?

The author agree the correct Reference is 51 in the RV.

  1. Castellana et al. assessed the prevalence of malignancy in each EU-TIRADS, class 5 compared to classes 2, 3 and 4. The authors found that the prevalence of malignancy was 16% in class 2, 5.5% in 3, 20.6% in 4 and 83.3% in 5 [82].The studies included in that review show malignancy risks for thyroid carcinoma around 50% - which is in no way comparable to primary care where this risk is far below 1%. For this reason the EU-TIRADS greatly overestimates malignancy risks – should be discussed. Bayes theorem.

The author agree. This sentence is added page 8 line 21“ However, the risk of malignancy is greater in highly specialized centers than in primary care centers. This is linked to the fact that selected patients come to highly specialized centers. This explains why in EU-TIRADS there is an overestimation of the risk of malignancy.

  1. The ATA and ETA guide-lines recommend US guided FNA in subcentimetric nodules only when suspicious fea-tures are present– somewhat misleading, since the US signs are not meant in ATA, but factors like suspicion of lymph node metastases, ETE and so on.

The author agree the ATA recommend US guide FNA in nodule with  Suspicious US feature > 1 cm. The ATA  and ACR TIRADs state that thyroid nodules <1 cm in size do not need to be evaluated with FNA , unless aggressive features  such as lymph node metastases, distant metastases and extrathyroidal extension are found.  This is different with respect Japan and Korean recommendation of performing FNA in  nodules >0,5 cm with highly suspicious US features.

Parts that have been modified are highlighted in yellow characters.

Authors hope that the manuscript is now acceptable for publication in Cancers

Round 2

Reviewer 1 Report

Although the revised manuscript has been improved, some errors in the cited references still exist. There are several comments and recommendations.

  1. Cited references

: The cited references should be correct especially in the review articles.

I strongly recommend to check the appropriateness of all cited references again throughout the manuscript including all tables.

Some of detected errors in cited references are as following.

- The reference 20 is missing in the reference list.

- P2, “Several endocrine societies have developed various US-based guidelines and recommendations for managing thyroid nodules [16-22].” As I commented before, all cited references are not proposals by endocrine societies. Please consider using “several researchers and professional societies” instead of “endocrine societies” if the cited references are not changed.

- P2. Shape “In a larger multicenter study, a taller-than-wide shape was shown to be highly suggestive of malignancy with a specificity of 89% and a positive predictive value of 86% [33].” The cited reference 33 in revised manuscript is “Hatabu, H.; Kasagi, K.; Yamamoto, K.; Iida, Y.; Misaki, T.; Hidaka, A.; et al. Cystic papillary carcinoma of the thyroid gland: a new sonographic sign. Clin Radiol 1991, 43:121-124.” Please check again if this cited reference is correct.

- The cited reference 83 in a foot note of Table 1 is not correct (reference 86 should be cited).

  1. Authors suggest that US-guided FNA is suggested for intermediate and high-risk nodules in ABSTRACT, and also described “All the guidelines agree that US guided FNA should be advised in high and intermediate risk category nodules, and not the low-risk category, such as cystic or spongiform nodules.” (P8).

- This proposal may be confusing to readers.

Although most guidelines agree that FNA is not routinely recommended for cystic or spongiform nodules with a very high specificity for benignity, the definition of “low-risk category” is not the same according to the societies’ guidelines.

Authors proposed 3 risk categories (Table 1). Current many US guidelines (EU-TIRADS, ATA, K-TIRADS, AACE/ACE/AME) recommend FNA for some nodules classified as low risk category if the nodule size is large (usually > 2 cm); for example, partially cystic ovoid nodules without intermediate or high risk US features based on definitions described in Table 1.  Although most of partially cystic isoechoic ovoid nodules without suspicious features are benign, follicular lesion malignant tumors such as follicular thyroid cancer, hurthle cell carcinoma, encapsulated FVPTC may show this US pattern. Therefore, majority of current guidelines except ACR TI-RADS recommend FNA for these nodules if the nodule size is large.

- Some nodules are not classified by the 3 risk categories proposed by authors (table 1). These unclassified nodules include partially cystic iso-/hyperechoic nodules without complete halo, solid iso-/hyperechoic nodules with or without complete halo among nodules without features of intermediate or high risk category.

All nodules should be basically classified by a risk stratification system to be widely used at clinical practice.

- These issues mentioned above should be considered in this review article.

  1. I recommend to use the recently revised 2021 K-TIRADS instead of 2016 K-TIRADS in table 2, which would be appropriate for the updated review article.

Reviewer 2 Report

Some additional information has been given to the statistical background of classification systems. A couple of sentences are not readily understandable, so the manuscript should be checked for phrasing.
